# Practice of people towards COVID-19 infection prevention strategies in Benishangul Gumuz Region, North–West Ethiopia: Multilevel analysis

**Muluwas Amentie Zelka**[1☯]*, **Melkamu Senbeta Jimma**[2☯], **Paulos Jaleta Wondashu**[2☯], **Atnafu Morka Aldo**[3☯], **Nigatu Disassa Abshoko**[4☯], **Dula Ayana Sadi**[1☯], **Wagari Kelbessa Gibina**[1☯]

1 Department of Public Health, College of Health Sciences, Assosa University, Assosa, Ethiopia,
2 Department of Nursing, College of Health Sciences, Assosa University, Assosa, Ethiopia, 3 Department of Geography and Environmental Studies, College of Social Sciences and Humanities, Assosa University, Assosa, Ethiopia, 4 Department of Animal Health, Colleges of Agriculture, Assosa University, Assosa, Ethiopia

☯ These authors contributed equally to this work.
* muluwas12@gmail.com

**Data Availability Statement:** All relevant data are within the paper and its Supporting Information files.

## Abstract

### Introduction

Coronavirus 2019 (COVID– 19) is an acute respiratory viral infectious disease in human being caused by RNA virus that belonged to the family of corona virus. The incidence of this disease was growing exponentially and affects millions of the world population that leads to expose thousands of peoples for death. Thus, this study was targeted to assess the practice of people on COVID-19 infections prevention strategies in the region.

### Methods

A community based cross sectional study design was conducted in Benishangul Gumuz Region from May 25 –June 15, 2020. Multistage sampling technique was carried out to select 21 kebeles/ketena and 422 study participants. Data were collected by face to face interview using semi-structured questionnaires. The collected data were entered, cleaned and analyzed using STATA software version 14.0. Descriptive, bi-variable and multivariable multilevel models were applied. Variables with p value < 0.25 in bi-variable analysis were selected as candidates for multivariable analysis. Finally, the variables with p-value < 0.5 were considered as statistically significant, then variables with odds ratio, 95% CI were used to interpret the effect of association.

### Results

The magnitude of good practice on prevention strategies of COVID– 19 infections was 62.1%. The most frequently practiced prevention strategies for COVID-19 infections were hand washing with water and soap (80.7%), alcohol-based hand rub (68.8%), maintaining

**Funding:** The author(s) received no specific funding for this work.

**Competing interests:** The authors have declared that no competing interests exist.

social/physical distance (74.2%) and avoiding touching eyes. Individual and community level factors that affecting practice of COVID– 19 infection prevention strategies were discovered. Hence, community level factor was place of origin (AOR = 0.1; 95%CI: 0.03, 0.35) whereas individual level factors were able to read and write (AOR = 0.18; 95%CI: 0.04, 0.81) and being merchant (AOR = 2.07; 95%CI: 1.01, 4.28).

## Conclusion

The level of practice of community towards COVID-19 infections prevention strategies were low as compared with the expected outcome. Individual and community level factors were identified. This implies that social mobilization and community engagement was not effective. Thus, designing appropriate strategies to improve of practice prevention strategies are strongly recommend.

## Introduction

Coronavirus 2019 (COVID– 19) is an acute respiratory viral infectious disease in human being caused by RNA virus that belonged to the family of corona virus, transmitted by respiratory and fomites with the incubation period from 2 to 14 days [1]. It remains one of the leading causes of morbidity and mortality around the world that challenging both developed and least developed countries [2]. The incidence of this disease was growing exponentially and affects millions of the world population exposing millions of peoples for death. Since December 2019, coronavirus disease (COVID-19) has become a pandemic, and patients with confirmed disease and those who are infected but asymptomatic are the main sources of infection [3].

Globally, on March 2021, there have been 124,535,520 confirmed cases and 2,738,876 deaths reported to WHO [4]. However, the practice and behaviors of people play a major role in the prevention and control of infectious diseases [5]. Adherence to WHO recommendation is likely to be influenced by the public practice of prevention strategies of COVID-19 infections. Evidences showed that public knowledge is an important mechanism in tackling pandemics of infectious diseases [6, 7]. Assessing community practices helps to identify attributes that influence the public in adopting healthy practices [8].

Despite many interventions declared by World Health Organization to reduce the spread of COVID- 19 infections, still now disease is spreading to different countries including Ethiopia. This might be because of having of poor practice of COVID-infection prevention strategies. The risk could be greater in developing region in Ethiopia such as Benishangul Gumuz Region with limited health facility, because it is situated at boarder of Sudan, have multiple point of entry, inadequate infrastructure, poor health care systems to provide required support.

Therefore, this study aims to assess the practice of community on COVID-19 infection in Benishangul Gumuz Region. As far as our search indicate that this research work was the only study conducted in Benishangul Gumuz Region on assessing the practice of community on COVID-19 infection. And the finding will help as a baseline for researchers, policymakers, regional government, public health officials, project implementers, clinicians and other stakeholders for doing better on prevention, preparedness and response for the COVID-19 outbreak in the Region.

## Methods and materials

### Study setting

Benishangul Gumuz Regional State is one of the eleven Federal Regions of Ethiopia situated in Northwest part of the country. It shares boundaries in the northern and northeastern with Amhara region, in the East with Oromia region, in the South with Gambela region and in the West with Republic of Sudan [9]. The total population of the region is 1.127 million. The region is located $9^0$35 N to $11^0$39 N latitudinally and longitudinally, from $34^0$20E to $36^0$30 E [10]. The elevation of the region ranges from 580 to 2,731metere above sea level. Three fourth of the region areas are lowland (kola) whereas 24% of areas are Woynadega and only 1% is highland (Dega). Administratively, the region is divided in to three zones namely Metekel, Assossa and Kamashi and Mao-Komo Special Woreda.

### Study design and period

A community based cross sectional study design was employed from June 10 to July 25, 2020 that aimed to assess the level of practice COVID– 19 infection prevention strategies and also determine individual and community level factors that affecting the practice level.

### Source population and study participants

Source of populations were all population whose age greater than 18 years old and permanent resident in the region and study population were randomly selected people.

### Variables and measurement

**Dependent variable.** Level of practice on COVID– 19 infection prevention strategies.

**Independent variables.** Socio-demographic, knowledge, attitude, community perception, individual perception and thinking, income and information on COVID– 19 infections.

### Inclusion and exclusion criteria

Community members whose age were greater or equal to 18 years old, volunteers and signed consent form were included but those who have mental, hearing, speaking or other severe medical problems were excluded in the study.

### Sample size determination and sampling technique

Single population proportion formula was used to determine sample size. The sample size was calculated using STATA software version 14 by considering the following assumption: proportion of practice of COVID-19 infections prevention as 50% (p = 0.5) because this infection is new and no scientifically known study has been conducted, with 95% level of confidence (two-sided α = 0.05) and considering 10% of non-response rate. Then, the final sample size was 422 study subjects.

Since this study conducted in the regional level, multistage sampling technique was used to identify the study participants. At the first stage, the region was stratified into Districts so called "Woredas" (20 "Woredas") and three town administration cities with one special Woreda. Then, six districts were selected from the 20 districts and one town administration was selected from three town administration cities and one special Woreda by using simple random sampling technique. Among the selected districts/town, 21 bebeles/ketenes were selected at the second stage using simple random sampling technique. At third stage, 422 household were selected using simple random sampling technique. Hence, household was considered as

final sampling unit (FSU). If more than one eligible people existed in the household, one individual could be selected using lottery method. Finally, with in one household, one individual was interviewed.

## Data collection and quality assurance

Data were collected using pre-tested semi-structured questionnaires. The research instruments were designed in English and translated to Amharic language and then back translated to English to ensure validity of the instruments. Then after, four day training was offered for 21 data collectors and seven BSc Nurse Holders were recruited as supervises. Following these, pretest was done on 5% of the total sample size in Assosa town administration woreda one ketene one. Before actual data collection, the necessary modification and amendment was made. During data collection, principal and co-investigators were acting as overall supervisor. Moreover, close supervision and daily checking of the questionnaires were made to ensure completeness and consistencies of the data collected.

## Data management and analysis

Data were coded and entered into Epi-info software and exported into STATA software version 14.0 for editing, cleaning and analyzing the information. Descriptive statistics were performed for all variables. The association of each factors was tested using $x^2$, z-test and t-test. Binary bi-variable logistic regression was performed and those variables with $p < 0.25$ were considered as candidates for the multivariable multilevel regression model. The strength of association was interpreted using AOR with 95% confidence interval. The criterion for statistical significance was set at 0.05. Specifically, mixed effect model was applied to determine individual and community level factors. The multi-collinearity effect between the independent variables was tested using variance inflation factors (VIF $>10$) and interaction effect at $p \geq 0.1$ was assessed. Since all variables included in the model had no multi-collinearity and interaction effect.

## Ethical approval and consideration

The research ethical issue was reviewed and approved by the research review senate stand committee of Assosa University. Then, ethical approval letter was obtained from Institutional Review Board (IRB) of Assosa University. Furthermore, permission letter was obtained from Benishangul Gumuz Regional State Health Bureau and also formal permission letter was received from all respective local administrators. Lastly, written informed consent was obtained from each respondent. Issues of confidentiality were maintained by removing any identifiers from the questionnaire.

## Results

### Socio–demographic characteristic of the study subject

Among the study subject included in the analysis, 209 (50.5%) were urban residents. Majority, 268(64.7%) belonged to the age group of 18–30 years with a mean (±SD) of 29.81±9.95. Regarding ethnic group, Amhara 118(28.5%) followed by Shinasha 74(17.9%). The majority of religion of the respondents was Orthodox Christian 228 (55.1%) followed by Muslim 121 (29.2%). More than half 273(65.9%) of the study subject were in a marital union. Regarding their educational background, 135(32.6%) of the respondents completed primary school (*Grade 1–8*) followed by high school (*Grade 9–12*) completed 95(22.9%). In line with occupational status, 114(27.5%) of the respondents were merchant followed by farmer 75 (18.1%) (Table 1).

**Table 1. Socio- demographic characteristic of the respondents in Benishangul Gumuz Region, Assosa, Ethiopia 2020.**

| Variables | Response category | No (%) |
|---|---|---|
| Place of resident | Urban | 209(50.5%) |
| | Rural | 205 (49.5%) |
| Gender of respondent | Male | 213(51.4%) |
| | Female | 201(48.6%) |
| Age of the respondent | 18–30 years old | 268(64.7%) |
| | 31–45 years old | 112(27.1%) |
| | ≥ 46 years old | 34(8.2%) |
| Religions | Orthodox | 228(55.1%) |
| | Muslim | 121(29.2%) |
| | Protestant | 34(8.2%) |
| | Catholic | 8(1.9%) |
| | Traditional belief | 5(1.2%) |
| | Others* | 18(4.3%) |
| Ethnic group | Berta | 73(17.6%) |
| | Amhara | 118(28.5%) |
| | Oromo | 71(17.1%) |
| | Gumuz | 58(14.0%) |
| | Shinasha | 74(17.9%) |
| | Others** | 20(4.8%) |
| Marital status | Married | 273(65.9%) |
| | Single | 115(27.8%) |
| | Divorced | 14(3.4%) |
| | Widowed | 12(2.9%) |
| Educational status | Unable to read and write | 64(15.5%) |
| | Read and write (informal edu.) | 14(3.4%) |
| | Primary school (Grade 1–8) | 135(32.6%) |
| | High school (Grade 1–8) | 95(22.9%) |
| | Diploma (Level I–IV) | 59(14.3%) |
| | First Degree (BSc/BA) | 40(9.7%) |
| | Master degree and above | 7(1.7%) |
| Occupational status | Farmer | 75(18.1%) |
| | Merchant | 114(27.5%) |
| | House wife | 53(12.8%) |
| | Governmental employee | 71(17.1%) |
| | Private employee | 24(5.8%) |
| | Student | 65(15.7%) |
| | Others*** | 12(2.9%) |
| **Total** | | **414(100%)** |

## Magnitude of practice of COVID-19 prevention strategies

**Source of information and it's seeking modality.** Source of information is crucial mechanism in infectious diseases prevention strategies. Not only the source of information but also the creditability and trustfulness of the source are mandatory in diseases prevention strategies. In line with these, 410 (99%) of the respondents did heard about COVID– 19 disease. Regarding the source of information; 278(67.1%) of the respondents received information from Television (TV) and 173(41.8%) from health workers (Table 2).

**Table 2. Source of information and seeking modalities on COVID-19 infection in Benishangul Gumuz Region, Ethiopia, 2020.**

|  | Responses of the respondents | Frequency | Percent |
|---|---|---|---|
| Have you heard about COVID– 19 disease | Yes | 410 | 99.0 |
|  | No | 4 | 1.0 |
| Source of information | Television (TV) | 278 | 67.1 |
|  | Radio | 195 | 47.1 |
|  | Friends/peer/Neighbors | 65 | 15.7 |
|  | Health workers | 173 | 41.8 |
|  | Community and religious leaders | 82 | 19.8 |
|  | Internet and social media | 69 | 16.7 |
|  | Others | 3 | 0.7 |
| Which source do you trust | Television | 266 | 64.3 |
|  | Radio | 166 | 40.1 |
|  | Friend/peers/neighbors | 19 | 4.6 |
|  | Health workers | 154 | 37.2 |
|  | Community and religious leader | 47 | 11.4 |
|  | Internet and social media | 22 | 5.3 |
|  | Others | 4 | 1.0 |
| Do you know call center service number in the region | Yes | 55 | 13.3 |
|  | No | 359 | 86.7 |
| The region call center service number to seek information about *COVID– 19* | 6016 | 36 | 8.7 |
|  | 8335 | 12 | 2.9 |
|  | 8035 | 2 | 0.5 |
|  | 444 | 2 | 0.5 |
|  | 806 | 1 | 0.2 |
|  | 5663 | 1 | 0.2 |
|  | 8535 | 1 | 0.2 |
|  | **Total** | **55(100%)** | **55(100%)** |

**Practice of people on COVID– 19 infection prevention strategies.** Among the parameters of COVID-19 infections prevention strategies, peoples practiced to protect themselves from spreading of COVID-19 infections are shown as follow. Of the respondents, 334(80.7%) practiced hand washing frequently with soap and water to kills the virus whereas 285(68.8%) practiced hand washing frequently with alcohol-based hand rub to kill the virus. Meanwhile, 307(74.2%) maintained keeping social/physical distance at least 2 meters whereas 318(76.8%) avoided touching eyes, nose and mouth to prevent the risk of infection.

In other view, 308(74.4%) of the respondents believed that isolation and treatment of people who are infected with the COVID-19 diseases is effective ways to reduce the spread of the virus while 342(82.6%) followed strictly the advice given by healthcare providers to reduce the chance of acquiring COVID-19 infections. In addition, 293(70.8%) of the respondents followed the recommended hand washing practices to prevent themselves from the diseases. However, it is only 326(78.7%) of the respondents that had the resource to wash their hands frequently with water and soup. The others that protected themselves from crowed places and close contact with anyone were 274(66.2%). Of the respondents, 216(52.2%) were wear a nose mask when they go out during the COVID-19 pandemic whereas 164(39.6%) of them carried sanitizer. On the other hand, 210(50.7%) supported/preferred/ that if lockdown policy imposed in Ethiopia to control COVID-19 infections.

Regarding to handling system of domestic animals, 117(28.3%) of the respondents had close intimation with domestic animals. The way of domestic animal management system was

assessed in which 135(51.3%) practiced intensive while 43(46.3%) practiced semi–intensive way. Moreover, 128(41.4%) of the respondents practiced hand washing before handling/feeding the animals whereas 139(47.4%) practiced hand washing after handling/feeding the animals. To this end, 124(30.0%) were used raw animal product for their household consumption.

The overall practice score of the respondents on COVID-19 infection prevention strategies was assessed by using multi-dimensional practices measurement indicators towards COVID–19 infections prevention strategies. Hence, the practice score was measured by using a composite variable of twenty-five items of the practice questions and the overall practice score was generated by PCA analysis. Then, respondents who scored above or equal to the means score were rated as having good practice on COVID-19 prevention strategies. Based on this, 257 (62.1%) were found to have good practice on COVID-19 infections prevention strategies with a mean ± SD of 16.86 ± 4.9 (Table 3).

**Parameter coefficients and test of goodness-of-fit of the multilevel models.** The individual and community level determinant factors affecting the respondents on practice COVID– 19 infection preventions strategies were detected by using statistical model such as multilevel logistic regression model. Before running the full model, ICC ($\rho$) was calculated in the empty model and full model for the outcome.

Thus, level of practice of the respondents on COVID– 19 infections prevention strategies, ICC ($\rho$) was calculated in the empty model and it was found to be 0.217 indicating that 21.7% of the variation was contributed by cluster variations. The test of the preference of log likelihood Vs logistic regression was also statistically significant (P<0.01). Then, the full model was run by including both the cluster level and individual level variables and the ICC ($\rho$) was increased to 0.238. Again this indicated that 23.8% of the variation was attributed to cluster level variables. The preference of log likelihood verses logistic regression was statistically significant (P = 0.02). Hence, this is suggesting that the appropriate model used to identify factors for this outcome is multilevel regression model (Table 4).

## Factors affecting practice of COVID– 19 infection prevention strategies

In the bi-variable analysis, different determinant factors such as place of resident, origin of place (currently they live), religion, marital status, educational status, occupational status, knowledge status, people's thinking level, individual perception, self-attitude and community perception have effect on prevention strategies of COVID– 19 infections.

After adjusting for confounders' effect in the regression model, among the cluster level variables, origin of place was found to have statistically significant association with practice of prevention strategies on COVID-19 infections. The odds of good practicing COVID– 19 infections prevention strategies among respondents whose place of origin that live in district town (AOR = 0.1; 95%CI: 0.03, 0.35) were 90% lower than among respondents reside in regional town.

Among the socio-demographic and economic characteristics that were considered as individual level (level– 1) factors, educational status was found to have statistically significant association with practice status of respondents on COVID-9 infections prevention strategies. The odds of having good practice in COVID– 19 infection strategies among respondents who able to read and write in educational status (AOR = 0.18; 95%CI: 0.04, 0.81) were 72% lower than among respondents who had not attended any formal education. Moreover, the odds of having good practice in COVID– 19 infection strategies among the participants whose occupational status as merchant (AOR = 2.07; 95%CI: 1.01, 4.28) was two times higher than participant whose occupational status as farmers (Table 5).

**Table 3. Practice of people on COVID– 19 prevention strategies in Benishangul Gumuz Region, Assosa, Ethiopia, 2020.**

| Measurement indicators | Responses | Number | Percent |
|---|---|---|---|
| Do you wash hands frequently with soap and water to kills the virus that causes COVID-19? | Yes | 334 | 80.7 |
| | No | 80 | 19.3 |
| Do you wash hands frequently with alcohol-based hand rub kill the virus that causes COVID-19? | Yes | 285 | 68.8 |
| | No | 129 | 31.2 |
| Do you maintain social/physical distance at least 2 meters that can prevent risk of infection with COVID-19? | Yes | 307 | 74.2 |
| | No | 107 | 25.8 |
| Do you avoid touching eyes, nose and mouth prevent infection with COVID-19? | Yes | 318 | 76.8 |
| | No | 96 | 23.2 |
| Do you cover your cough/sneezing using the bend of your elbow or a tissue prevent spread of COVID-19? | Yes | 290 | 70.0 |
| | No | 124 | 30.0 |
| Do you avoid crowed places and close contact with anyone prevent risk of infection with COVID-19? | Yes | 311 | 75.1 |
| | No | 103 | 24.9 |
| Do you stay at home help to prevent infections with COVID-19? | Yes | 260 | 62.8 |
| | No | 154 | 37.2 |
| Do you accept/practice that isolation and treatment of people who are infected with the COVID-19 are effective ways to reduce the spread of the virus | Yes | 308 | 74.4 |
| | No | 106 | 25.6 |
| Do you follow advice given by your healthcare provider can reduce the chance of acquiring COVID-19 | Yes | 342 | 82.6 |
| | No | 72 | 17.4 |
| Do you follow recommended hand washing practices to prevent myself from COVID-19? | Yes | 293 | 70.8 |
| | No | 121 | 29.2 |
| Do you have the resource to wash your hands frequently with water and soup to prevent yourself from COVID-19? | Yes | 326 | 78.7 |
| | No | 88 | 21.3 |
| Do you protect yourself from crowed places and close contact with anyone to protect yourself from COVID-19? | Yes | 274 | 66.2 |
| | No | 140 | 33.8 |
| Do you wear a nose mask when you go out during the COVID-19 pandemic? | Yes | 216 | 52.2 |
| | No | 198 | 47.8 |
| Do you carry sanitizer during the COVID-19 pandemic? | Yes | 164 | 39.6 |
| | No | 250 | 60.4 |
| Do you support if lockdown imposed in Ethiopia to control COVID-19? | Yes | 210 | 50.7 |
| | No | 204 | 50.7 |
| Ethiopians greets each other by handshake followed by shoulder hit for man, cheeks touch for women, do you prefer this way of greetings in this COVID-19 pandemic? | Yes | 156 | 37.7 |
| | No | 258 | 62.3 |
| Do you limit your movement during the COVID– 19 pandemics? | Yes | 285 | 68.8 |
| | No | 129 | 31.2 |
| Do you have close intimation with domestic animals? | Yes | 117 | 28.3 |
| | No | 297 | 71.7 |
| The way of managing domestic animals? | Intensive | 135 | 51.3 |
| | Semi-intensive | 43 | 46.3 |
| | Extensive | 85 | 32.3 |
| Do you practice hand washing before handling/feeding the animals | Yes | 128 | 41.4 |
| | No | 181 | 58.6 |
| Do you practice hand washing after handling/feeding the animals | Yes | 139 | 47.4 |
| | No | 154 | 52.6 |
| Do you use raw animal product for your household consumption? | Yes | 124 | 30.0 |
| | No | 290 | 70.0 |

*(Continued)*

**Table 3.**  (Continued)

| Measurement indicators | Responses | Number | Percent |
|---|---|---|---|
| If I have symptoms of COVID-19, do you isolate yourself from your family? | Yes | 367 | 88.6 |
| | No | 47 | 11.4 |
| If I have symptoms of COVID-19, do you isolate yourself from your community? | Yes | 368 | 88.9 |
| | No | 46 | 11.1 |
| If I have symptoms of COVID-19, do you report to quarantine site for further investigation and treatment? | Yes | 365 | 88.2 |
| | No | 49 | 11.8 |
| Do you support the traffic blockage and limit the number of travelers in region to control COVID-19? | Yes | 247 | 59.7 |
| | No | 167 | 40.3 |
| *Overall practice status of COVID– 19 prevention strategies* | *Good practice** | *257* | *62.1* |
| | *Poor practice*** | *157* | *37.9* |
| | *Mean ± SD = 16.86 ± 4.9* | | |

* indicates that the overall score of the measurement indicators greater than or equal to mean value.

** indicates that the overall score of the measurement indicators less than mean value.

# Discussion

## Practice towards COVID– 19 infection prevention strategies

This study found that peoples practiced to COVID– 19 infections prevention strategies such that 62.1% of the people were adhered to good practice on prevention modality of the diseases. This finding was lower than study conducted in Chine which revealed that the average score of good practice found to be 96.8% in COVID-19 prevention practices: wearing face masks and maintaining hand hygiene should be prioritized to prevent its transmission. This is due to Benishangul Gumuz Region is one of the remote and underprivileged regions in Ethiopia and also Ethiopia is one of the poorest countries in the world. Moreover, there was poor accessibility of water and personal protective equipment in the region. As a result, the level of practice COVID– 19 prevention strategies is lower than developed countries [11]. In factor, people whose occupational status is "*merchants*" and good community perception ultimately had good practice. However, this finding was more or less consistent with evidence from Bangladesh reported 55.1% had more frequent practicing COVID– 19 infections prevention strategies [12]. Moreover, this finding supported by other study conducted in Sierra Lion (74.6%), Amhara region (62%), Northern Ethiopia (67%), China (96.8%), Bangladesh (55.1%), North-central Nigeria (90.46%) and Northwest Ethiopia (52.7%) were frequently practiced COVID–

**Table 4.  Parameter coefficients and test of goodness-of-fit of the mixed-effects multilevel models, Benishangul Gumuz Region, Ethiopia 2020.**

| Models | Fixed intercept -cons(95%CI) | Random effect as Level-2 variance var (-cons (95%CI)) | Intra-class Correlation Coefficient: ICC(ρ) | Log likelihood (LR)-deviance | Significance of LR test Vs Logistic regression (P-value) |
|---|---|---|---|---|---|
| **Practice status*** | | | | | |
| *Empty model* | 0.6(0.45, 0.72) | 0.05(0.03, 0.12) | 0.217 = 21.7% | - 258.9 | P < 0.01 |
| *Full model* | 0.64(0.24, 1.04) | 0.05(0.02, 0.11) | 0.238 = 23.8% | - 255.6 | P = 0.02 |

P value less than 0.05 is statistically significant and the data fit for multilevel model.

* Multilevel regression model applied to measure the effect of factors on this outcome.

**Table 5. Multilevel analysis of factors effecting practice of COVID 19 infections prevention strategies, among respondents, Benishangul Gumuz Region, Ethiopia 2020.**

| Factors | Practice status | | Crude OR (95%CI) | Adjusted OR (95%CI) |
|---|---|---|---|---|
| | *Poor practices* | *Good practices* | | |
| | *N (%)* | *N (%)* | | |
| **Level-2: Higher level variables** | | | | |
| Residents: Urban | 65(31.1) | 144(68.9) | 1 | 1 |
| Rural | 92(44.9) | 113(55.1) | *0.55(0.37, 0.83)* | 0.29(0.02, 3.69) |
| Origin of place: Regional town | 4(8.0) | 46(92.0) | 1 | 1 |
| District town | 62(38.8) | 98(61.2) | *0.14(0.05, 0.4)* | *0.1(0.03, 0.35)* |
| Rural kebele | 91(44.6) | 113(55.4) | *0.11(0.04, 0.31)* | 0.35(0.02, 5.75) |
| **Level-1: Lower-level variables (Individual level factors)** | | | | |
| Religion: Orthodox | 79(34.6) | 149(65.4) | 1 | 1 |
| Muslim | 45(37.2) | 76(62.8) | 0.89(0.57, 1.42) | 0.99(0.58, 1.67) |
| Others | 33(50.8) | 32(49.2) | *0.51(0.29, 0.89)* | 0.65(0.33, 1.26) |
| Marital status: Married | 99(36.3) | 174(63.7) | 1 | 1 |
| Single | 42(36.5) | 73(63.5) | 0.99(0.63, 1.56) | 0.93(0.53, 1.64) |
| Others | 16(61.5) | 10(38.5) | *0.36(0.16, 0.81)* | 0.5(0.19, 1.33) |
| Educational status: Illiterate | 37(57.8) | 27(42.2) | 1 | 1 |
| Read and write | 10(71.4) | 4(28.6) | 0.55(0.16, 1.94) | *0.18(0.04, 0.81)* |
| Primary school | 48(35.6) | 87(64.4) | *2.48(1.35, 4.56)* | 1.26(0.61, 2.59) |
| High school | 32(33.7) | 63(66.3) | *2.67(1.4, 5.19)* | 1.02(0.43, 2.41) |
| Diploma | 16(27.1) | 43(72.9) | *3.68(1.73, 7.87)* | 1.79(0.69, 4.67) |
| Degree and above | 14(29.8) | 33(70.2) | *3.23(1.45, 7.17)* | 1.79(0.23, 5.16) |
| Occupational status: Farmer | 39(52.0) | 36(48.0) | 1 | 1 |
| Merchant | 35(30.7) | 79(69.3) | *2.45(1.34, 4.47)* | *2.07(1.01, 4.28)* |
| House wife | 22(41.5) | 31(58.5) | 1.53(0.75, 3.1) | 0.83(0.33, 2.05) |
| Gov't employee | 23(32.4) | 48(67.6) | *2.26(1.15, 4.43)* | 0.87(0.36, 2.14) |
| Private employee | 8(33.3) | 16(66.7) | 2.17(0.83, 5.67) | 0.78(0.25, 2.49) |
| Students | 24(36.9) | 41(63.1) | 1.85(0.94, 3.64) | 1.32(0.56, 3.13) |
| Others | 6(50.0) | 6(50.0) | 1.08(0.32, 3.67) | 0.98(0.26, 3.73) |

19 infections prevention strategies [11–17]. However, the proportion of the entire practices level was not as high in contrast to the contagiousness natures of the virus.

In the absence of practicing COVID– 19 infections prevention strategies, the entire public is at high to contract with COVID– 19 infections which fostering the spreading of virus throughout the world. Specifically, the common COVID– 19 infection prevention that the participants were practicing in study area was washing hand. In this regard, 80.7% of the participants were practicing washing hand with water and soap. Among the study participants, 70.8% were following the recommended hand washing frequency with water and soap. In addition, 74.2%, 70% and 75% were maintaining social/physical distance, covering their nose and mouth while cough/sneezing using bend of their elbow or a tissue and avoiding crowed places/close contact with any one respectively. Beside, 62.8% were staying at home, 74.4% were accepting and practicing isolation and treatment of people who are infected with COVID– 19 infections, 52.2% were wearing the nose mask while they go outside the home and 39.6% were carrying sanitizers or alcohol to clear or rub their hand while they touch any things. Similarly, 68.8% of the study participants were accepting traffic blockage and limiting their movement to protect themselves from the pandemic disease. This finding consistent with the study conducted in Tigray region which revealed that 49.8% of the participants have gone to crowded

places in recent days and 46% of the people did not use a face mask when leaving home [18]. Other study in Bangladesh found to be 71% of the participants were washing their hands more frequently than ever and for an extended period [19].

Similarly, study in Saudi Arabia were found to be practicing social distancing (84.6%), frequently washing their hands with soap and water, for at least 40 seconds, especially after going to a public place, or after nose-blowing, coughing, or sneezing (73.08%) and avoided cultural behaviors, such as shaking hands (87.57%) [20]. In contrarily, study done in Tigray region revealed that 54.4% did not obey the preventive measures given by local health care authorities [18]. However, evidence in Southwest Ethiopia revealed that properly washing hands with soap and water (95.5%), not touching eye-nose-mouth with unwashed hands (92.7%), and avoiding crowded places (90.3%) were commonly known and practicing as COVID– 19 infections prevention strategies [21]. Moreover, Jimma University Medical Center visitors were predominantly engaged in frequently hand washing with water and soap (77.3%), stopped shaking hands while giving greeting (53.8%), avoiding physical proximity (33.6%) and limiting went to crowed places (33.2%) to protect themselves from COVID-19 [21].

Other study in Malaysia revealed that 51.2% of the total participants used face mask when leaving home [22]. It is contrary to studies done in Tanzania where 77%, and 80% avoided crowds and wore face masks [23]; in India, majority of participants took preventive measures [24]; in China 96.4% and 98% of study participants were avoiding crowded condition and used a face mask respectively [25]; in Nepal 94.9%, 88.2%, and 93.7% of participants were avoiding crowded condition/place, appropriately used a face mask, and took preventive measures respectively [26] and in Malaysia, 83.4% and 87.8% of the people were avoiding crowding condition and took precautionary measures respectively [22]. Moreover, study in North–central Nigeria revealed that following/respecting health recommendations, social distancing/avoiding crowd, avoiding handshakes and face kissing were some of the preventive measurements that the community practices to reduce community spread COVID-19 infections as reported by 90.2%, 78.8% and 74.4% respectively [15].

In this study, we found that 37.7% of the people were practicing and preferring Ethiopian greeting each other by handshake followed by shoulder hit for man and cheeks touch for women and 28.3% had close intimation with domestic animals. This evidence supported by study done in Tigeri region which revealed that 54.4% of participants did not follow the preventive measures given by local health care authorities [18]. Similarly, study done in Southwest Ethiopia revealed still avoidance of non-careful touching of face parts that are used for the entrance of the virus, wearing masks/cover while coughing/sneezing, use of sanitizers, and stay at home was very low (1.6–11.3%) [21].

This study explored that individual and community level factors associated with practices of preventive strategies on COVID– 19 infections. Consequently it portrays that being living in regional town, educational status and merchant in their occupation found to have good practices on COVID– 19 infection prevention strategies and also those communities who had good perception towards COVID– 19 infections would have good practice on COVID– 19 infections prevention modality. This finding supported by study done in Tigray region piloted that as go to more educated, they tend to practice preventive measures to COVID– 19 pandemics [18]. Other evidence document that socio-demographic factors associated with more frequently practice preventative measures such that people who were being students, having higher education, urban area residence, having more positive attitudes had more likely to had good practice on COVID– 19 infections preventative measurements. These determinants were fostering the level of practice on COVID– 19 infection prevention strategies [26, 27].

Similarly, study in Saudi Arabia showed that as 1% increase in knowledge score is associated an increase in practices scores, of 0.16%. This implies that the knowledge score of the

participants had positive association towards practicing of COVID– 19 infections preventions strategies [20].

Therefore, majority of the public adhered to good practices with respect to COVID-19 infection, because potentially they had good practice towards COVID– 19 infections prevention though not enough. However, shortage of protective equipment such as medical alcohol and sanitizer, lack of accessibility of water and soap, shortage of medical mask and chlorine-containing disinfectants were the challenges that the participants encountered. This result suggests that during infectious disease outbreaks, government departments should try their best to provide sufficient supplies of protective equipment so that the public can take protective actions.

## Conclusion and recommendations

The level of good practice towards COVID-19 infections prevention strategies were low as compared with the expected outcome. This is affected by different individual and community level factors. This implies that social mobilization and community engagement was not as such effective. Therefore, designing appropriate strategies at community levels to improve practice of community on COVID-19 infections prevention strategies are strongly recommended.

## Supporting information

**S1 File. List of study site (kebeles/ketenas) included in the study in Northwest Ethiopia, 2020.**
(DOCX)

**S2 File. Research instrument to assess the level of practice in COVID -19 infection prevention strategies, Benishangul Gumuz Region, Northwest, Ethiopia.**
(DOCX)

**S1 Dataset. Minimal dataset used to analyze for this manuscript only.**
(SAV)

## Acknowledgments

We would like to give our gratitude to Assosa University, Benishangul Gumuz Regional Health Bureau, Zonal Health Department and District Health Office. We thank all community members, data collectors and supervisors who were represented in this study.

## Author Contributions

**Conceptualization:** Muluwas Amentie Zelka, Melkamu Senbeta Jimma, Paulos Jaleta Wondashu, Atnafu Morka Aldo, Nigatu Disassa Abshoko.

**Data curation:** Muluwas Amentie Zelka, Melkamu Senbeta Jimma, Paulos Jaleta Wondashu, Atnafu Morka Aldo, Nigatu Disassa Abshoko, Dula Ayana Sadi, Wagari Kelbessa Gibina.

**Formal analysis:** Muluwas Amentie Zelka, Melkamu Senbeta Jimma, Paulos Jaleta Wondashu, Atnafu Morka Aldo, Nigatu Disassa Abshoko.

**Methodology:** Muluwas Amentie Zelka.

**Software:** Muluwas Amentie Zelka.

**Supervision:** Muluwas Amentie Zelka, Melkamu Senbeta Jimma, Paulos Jaleta Wondashu, Atnafu Morka Aldo, Dula Ayana Sadi, Wagari Kelbessa Gibina.

**Validation:** Muluwas Amentie Zelka, Paulos Jaleta Wondashu, Atnafu Morka Aldo, Nigatu Disassa Abshoko, Dula Ayana Sadi, Wagari Kelbessa Gibina.

**Visualization:** Muluwas Amentie Zelka, Melkamu Senbeta Jimma, Paulos Jaleta Wondashu, Atnafu Morka Aldo, Nigatu Disassa Abshoko, Wagari Kelbessa Gibina.

**Writing – original draft:** Muluwas Amentie Zelka, Melkamu Senbeta Jimma.

**Writing – review & editing:** Muluwas Amentie Zelka, Melkamu Senbeta Jimma, Paulos Jaleta Wondashu, Atnafu Morka Aldo, Nigatu Disassa Abshoko, Dula Ayana Sadi, Wagari Kelbessa Gibina.

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
