## [Decision Letter · Decision Letter 0]

19 May 2021

PONE-D-21-10875

Practice of people towards COVID-19 infection prevention in Benishangul Gumuz Region, North-West Ethiopia, 2020

PLOS ONE

Dear Dr. Zelka,

Thank you for submitting your manuscript to PLOS ONE. After careful consideration, we feel that it has merit but does not fully meet PLOS ONE’s publication criteria as it currently stands. Therefore, we invite you to submit a revised version of the manuscript that addresses the points raised during the review process.

We look forward to receiving your revised manuscript.

Kind regards,

Seyed Ehtesham Hasnain, Ph.D

Academic Editor

PLOS ONE

Additional Editor Comments:

Major Revision

Journal Requirements:

4. In your manuscript, please provide a summary table of participant demographics.

5. Please provide table 2 in the body of your manuscript.

6. Please provide a list of the names of the 21 Kebeles that were sampled from as a supplementary file.

7. Thank you for stating in the text of your manuscript "ethical approval was obtained from Institutional Review Board (IRB) of Assosa University" and "written informed consent was obtained from each respondent". Please also add this information to your ethics statement in the online submission form.

8. In your Data Availability statement, you have not specified where the minimal data set underlying the results described in your manuscript can be found. PLOS defines a study's minimal data set as the underlying data used to reach the conclusions drawn in the manuscript and any additional data required to replicate the reported study findings in their entirety. All PLOS journals require that the minimal data set be made fully available. For more information about our data policy, please see http://journals.plos.org/plosone/s/data-availability.

9. Please include a copy of Table 3 which you refer to in your text on page 8. Currently you have included Table 1 two times.

10. Please include captions for your Supporting Information files at the end of your manuscript, and update any in-text citations to match accordingly. Please see our Supporting Information guidelines for more information: http://journals.plos.org/plosone/s/supporting-information.

Reviewers' comments:

Reviewer's Responses to Questions

**Comments to the Author**

1. Is the manuscript technically sound, and do the data support the conclusions?

Reviewer #1: Partly

Reviewer #2: Partly

2. Has the statistical analysis been performed appropriately and rigorously? 

Reviewer #1: Yes

Reviewer #2: I Don't Know

3. Have the authors made all data underlying the findings in their manuscript fully available?

Reviewer #1: No

Reviewer #2: Yes

4. Is the manuscript presented in an intelligible fashion and written in standard English?

Reviewer #1: No

Reviewer #2: No

5. Review Comments to the Author

Reviewer #1: The authors have assessed the practice of community towards COVID-19 infections prevention strategies. This is a poorly designed study. This reviewer thinks the sample size considered in the present study is very low although the authors have nowhere mentioned the sample size. What were the questions in the questionnaire and what was the range of the knowledge score? The language used in the present study makes it even more difficult to understand the work done. The authors should consider a professional to improve the language and make the study understandable.

Reviewer #2: The authors have tried to explain the basic practices being followed at every corner around the world. There is no novelty and it is a broad overview of the practices. I suggest them to improve the manuscript significantly in terms of their data (add diagnosis or treatment related real time data) and writing format.

6. PLOS authors have the option to publish the peer review history of their article (what does this mean?). If published, this will include your full peer review and any attached files.

Reviewer #1: No

Reviewer #2: No

---

## [Author Response · Author response to Decision Letter 0]

8 Dec 2021

General reflection on comment: Really, we are very happy and satisfied with the comment given by the reviewers and editors. Based on the comments, we have significantly changed and improved our manuscript. Then, the specific reflection on the comments were described in the table form and attached in response to reviewer catalog.

Thanks again and again for your nice comment and suggestion

Best regard 

Muluwas Amentie

---

## [Editor Report · Decision Letter 1]

24 Jan 2022

Practice of people towards COVID-19 infection prevention strategies in Benishangul Gumuz Region, North –West Ethiopia: Multilevel Analysis

PONE-D-21-10875R1

Dear Dr. Zelka,

We’re pleased to inform you that your manuscript has been judged scientifically suitable for publication and will be formally accepted for publication once it meets all outstanding technical requirements.

Kind regards,

Seyed Ehtesham Hasnain

Section Editor

PLOS ONE

Additional Editor Comments (optional):

The manuscript was sent for major revision and Authors have modified the manuscript keeping in mind the comments of the Reviewers. I have gone through the revised manuscript and also the Author response to the comments of the Reviewers. Authors have included the socio-demographic table and results within the manuscript at page no.8. The sample sizes have been calculated following scientific procedures and Authors have used the single population proportion formula which scores the largest sample size compared with two population proportion formulas which applied for associated factors. This manuscript has been comprehensively revised by the Authors addressing the comments of both the reviewers. All other explanations provided by the Authors are quite satisfactory. I recommend this manuscript for publication.
---

## [Editor Report · Acceptance letter]

3 Feb 2022

PONE-D-21-10875R1 

 Practice of people towards COVID-19 infection prevention strategies in Benishangul Gumuz Region, North –West Ethiopia: Multilevel Analysis 

Dear Dr. Zelka:

I'm pleased to inform you that your manuscript has been deemed suitable for publication in PLOS ONE. Congratulations! Your manuscript is now with our production department. 

Kind regards, 

on behalf of

Prof. Seyed Ehtesham Hasnain 

Section Editor

PLOS ONE